# The impact of nutritional status in nivolumab-treated patients with advanced esophageal cancer

**Naoki Takegawa, Taku Hirabayashi, Shunta Tanaka, Michiko Nishikawa, Nagahiro Tokuyama, Takuya Mimura, Saeko Kushida, Hidetaka Tsumura, Yoshinobu Yamamoto, Ikuya Miki, Masahiro Tsuda** *

Department of Gastroenterology, Hyogo Cancer Center, Akashi, Hyogo, Japan

* b4hkgsntgm7@hyogo-cc.jp

**Data Availability Statement:** All relevant data are available. Please see http://dx.doi.org/10.6084/m9.figshare.22331341, and http://dx.doi.org/10.6084/m9.figshare.22331353.

## Abstract

Although phase III trials have reported improved overall survival in patients with advanced esophageal squamous cell carcinoma following treatment with nivolumab, as compared with chemotherapy (paclitaxel or docetaxel), the treatment was effective only in a limited number of patients. Therefore, the aim of this study is to determine whether there is a correlation between nutritional status (Glasgow prognostic score, prognostic nutritional index, and neutrophil-to-lymphocyte ratio) and prognosis of advanced esophageal cancer in patients treated with taxane or nivolumab therapy. The medical records of 35 patients who received taxane monotherapy (paclitaxel or docetaxel), for advanced esophageal cancer between October 2016 and November 2018 (taxane cohort) were reviewed. The clinical data of 37 patients who received nivolumab therapy between March 2020 and September 2021 (nivolumab cohort) were collected. The median overall survival was 9.1 months for the taxane cohort and 12.5 months for the nivolumab cohort. In the nivolumab cohort, patients with good nutritional status had significantly better median overall survival than those with poor nutritional status (18.1 vs. 7.6 months, respectively, $p = 0.009$, classified by prognostic nutritional index, 15.5 vs. 4.3 months, respectively, $p = 0.012$, classified by Glasgow prognostic score), whereas the prognosis of the patients treated with taxane therapy was less affected by the nutritional status. This suggests that the pretreatment nutritional status of patients with advanced esophageal cancer is a key factor for successful outcomes, especially for treatment with nivolumab.

## Introduction

Programmed cell death 1 (PD-1) immune checkpoint inhibitors have demonstrated promising treatment outcomes in multiple cancers. In a phase III study, nivolumab, a PD-1 immune checkpoint inhibitor, showed a survival benefit over taxane therapy in patients treated for advanced esophageal cancer (AEC) [1].

Nutritional assessment is an essential component in the evaluation of patients with gastrointestinal cancer since their clinical course can be complicated by undernutrition. In addition,

**Funding:** This study received no specific funding for this work.

**Competing interests:** The authors have declared that no competing interests exist.

nutritional status affects treatment outcome as well as the incidence of adverse events [2,3]; therefore, nutritional assessment of cancer patients should be considered according to the European Society for Clinical Nutrition and Metabolism (ESPEN) guidelines on enteral nutrition [4]. Several simple and reliable screening tools have been used to evaluate the nutritional status of cancer patients, of which the prognostic nutritional index (PNI), Glasgow prognostic score (GPS), and neutrophil-to-lymphocyte ratio (NLR) are commonly used. The PNI is calculated using the serum albumin concentration and total lymphocyte count in the peripheral blood. Albumin is important for the transport of nutrients and supports body metabolism. Lymphocytes and their effector cells are key constituents of anticancer immune responses [5]. PNI quantifies nutritional and immunological status in patients [6]. GPS, which can predict the postoperative outcomes of various cancers [7–9], is a combination of serum C-reactive protein (CRP) and albumin levels, which are biomarkers of systemic inflammatory response and nutritional status, respectively [10]. GPS can be used to assess nutritional status since inflammation in patients contributes to the development of cachexia which further worsens the nutritional status.

We have previously reported [11] that pretreatment-GPS was a predictor of outcomes in nivolumab-treated patients with advanced gastric cancer (AGC). NLR is calculated as the ratio of circulating blood neutrophils and lymphocytes. Neutrophils are a component of peritumoral inflammatory cell infiltrates, which show cancer cell activity. A high NLR is associated with worse clinical outcomes in esophageal cancer [12].

Based on our previous findings [11], we hypothesize that the pretreatment nutritional status of patients with AEC influences the treatment outcome of nivolumab therapy. To test this hypothesis, this study retrospectively investigated the relationship between the outcomes of nivolumab or taxane therapy and nutritional markers assessed at the time of treatment initiation.

## Patients and methods

### Study design and patients

The medical records of all patients with advanced or recurrent (stage IV) esophageal cancer at Hyogo Cancer Center Hospital, Hyogo, Japan, who received nivolumab monotherapy between March 2020 and September 2021, were retrospectively reviewed (nivolumab cohort). The consecutive patients with AEC who received taxane monotherapy (paclitaxel or docetaxel) were also analyzed (taxane cohort). To exclude the use of immune checkpoint inhibitors, the data were collected from patients treated with taxane therapy between October 2016 and November 2018 (before the approval of nivolumab for treatment of AEC). The eligibility criteria for patient inclusion were as follows: aged $\geq$ 20 years; histologically diagnosed esophageal cancer; measurable or evaluable lesions according to the Response Evaluation Criteria in Solid Tumors (RECIST) version 1.1; preserved Eastern Cooperative Oncology Group (ECOG) performance status (PS) of $\leq$ 2; no major organ dysfunction; no active multiple primary cancers; availability of clinical data of serum CRP, albumin levels, neutrophil count, and lymphocyte count (the latest data before administration); refractory or intolerant to fluoropyrimidine-based and platinum-based chemotherapy. Refractory disease is defined as cancer with progression during chemotherapy (including chemoradiation) or recurrence within 24 weeks after the last dose of chemotherapy if prior chemotherapy (including chemoradiation) resulted in a complete response or surgical resection. This study was approved by the ethics committee of Hyogo Cancer Center (approval number G-200, 2021). The ethics committee waived the requirement for informed consent, and all the data were analyzed anonymously.

## Treatment

Intravenous Nivolumab was administered at a flat dose of 240 mg for two weeks. Paclitaxel was administered intravenously at 100 mg/m$^2$ once per week for six weeks, followed by a week off. Docetaxel was administered intravenously at 60 mg/m$^2$ intravenously every three weeks. Treatment was continued until the occurrence of disease progression, unacceptable toxicity, or patient refusal. The patients were followed-up until March 1, 2023.

## Evaluation

Patient information was retrieved from the electronic medical records of the hospital for retrospective analysis. The data collected for this study included age, sex, ECOG PS, smoking status, histology, metastatic sites, number of prior therapies, PNI, GPS, and NLR. Progression-free survival (PFS) was defined as the time from the administration of the first dose of nivolumab or taxane to the occurrence of disease progression or death. Overall survival (OS) was defined as the time from the initiation of nivolumab or taxane treatment until death by any cause or from the last follow-up. Chemotherapy information included the best response, PFS, and OS. The overall response rate was assessed using RECIST version 1.1 [13]. The following markers, reflecting the nutritional status of the patient, were investigated: PNI, GPS, and NLR. These markers were assessed before the initiation of nivolumab or paclitaxel treatment. PNI was calculated using the formula [10 × albumin (g/dL) + 0.005 × absolute lymphocyte count/μL], and the cut-off value was set at 45 according to previous studies [14,15]. The GPS was calculated as a combination of CRP and albumin levels. Patients with increased CRP (> 1.0 mg/dL) and hypoalbuminemia (< 3.5 g/dL) were assigned a score of 2. Patients with only one abnormal value were assigned a score of 1. Patients with normal CRP and albumin levels were assigned a score of 0. NLR was calculated as the ratio of absolute neutrophil count/μL to absolute lymphocyte count/μL, and an NLR threshold of 5 was used [16,17].

Further, the patients were divided into two groups according to their nutritional status: good nutritional status (low GPS [GPS 0], high PNI [PNI ≥ 45], and low NLR [NLR < 5]) and poor nutritional status (high GPS [GPS 1 or 2], low PNI [PNI < 45], and high NLR [NLR ≥ 5]).

## Statistical analyses

The baseline characteristics were compared using the $\chi^2$ test or Fisher's exact test. Kaplan-Meier analyses of OS and PFS were performed based on these cut-off values, with differences between each pair of groups assessed using the log-rank test. The hazards ratio (HR) and 95% confidence interval (CI) were calculated using the univariate Cox proportional hazards model. All $p$ values were based on a two-sided hypothesis, with statistical significance set at p < 0.05. The software used was the Statistical Package for Social Sciences (SPSS, Chicago, IL, version 27).

# Results

## Patient characteristics

From October 2016 to November 2018, 39 patients underwent taxane monotherapy, of which 3 were excluded due to a prior administration of immune checkpoint inhibitors in previous clinical trials, and 1 patient was excluded due to the absence of pathological confirmation of cancer. A total of 35 patients were included in the taxane cohort. One patient remained untraceable after 7.4 months of starting paclitaxel therapy. A total of 37 patients received nivolumab monotherapy between March 2020 and September 2021 (nivolumab

**Fig 1. Flow diagram of the study.**

cohort). All patients treated with nivolumab were included in the analysis (Fig 1). The median follow-up time of the censored patients was 23.3 months in the nivolumab cohort. The baseline patient characteristics are summarized in Table 1. The median age of the patients was 64 years (range 46–81 years) in the taxane cohort and 67 years (range 46–84 years) in the nivolumab cohort. In both cohorts, the predominant histology was squamous cell carcinoma, followed by adenocarcinoma. More patients received two or more prior therapies in the nivolumab cohort than in the taxane cohort, which was statistically significant ($p = 0.013$). When assessed for PNI, GPS, and NLR status, the patients were equally distributed in both treatment groups.

## Treatment outcomes

By the end of the follow up period, 34 patients in the taxane cohort and 29 patients in the nivolumab cohort had died. All 35 patients in the taxane cohort, 34 patients in the nivolumab cohort showed disease progression, and 3 patients in the nivolumab cohort were undergoing treatment. The overall response was evaluated in 25 patients in the taxane cohort and 27 patients in the nivolumab cohort with measurable lesions. The overall response rate (ORR) was 20.0% (5/25) for taxane and 22.2% (6/27) for nivolumab (S1 Table); two patients treated with nivolumab showed complete response. The median PFS and OS in the study population were 2.5 months (95% CI: 1.3–3.7) and 9.1 months (95% CI: 5.0–13.2) in the taxane cohort, respectively (S1 Fig), and 2.3 months (95% CI: 1.6–3.1) and 12.5 months (95% CI: 5.3–19.8) in the nivolumab cohort, respectively (S2 Fig).

One patient received subsequent systemic chemotherapy with docetaxel in the taxane cohort, while 18 received paclitaxel, 1 received docetaxel, 1 patient received trifluridine/tipiracil, and 1 patient received chemoradiation therapy in the nivolumab cohort.

## The impact of nutritional status in the taxane cohort

Univariate analysis for OS showed that female patients lived significantly longer than male patients in the taxane cohort (Table 2). When stratified according to PNI, GPS, and NLR scores, patients with good nutritional status tended to have longer OS, although not statistically significant (HR = 1.326, $p = 0.437$ [classified by PNI]; HR = 1.831, $p = 0.105$ [classified by GPS]; and HR = 1.501, $p = 0.314$ [classified by NLR]). The Kaplan–Meier analysis showed no significant differences between PFS, OS, and nutritional status classified by PNI, GPS, or NLR in this cohort (Fig 2 and S3 Fig).

**Table 1. Patient characteristics.**

| | Taxane cohort (n = 35) | Nivolumab cohort (n = 37) | p value |
|---|---|---|---|
| Age in years (range) | 64 (46–81) | 67 (46–84) | 0.328 |
| Sex | | | 0.367 |
| Male | 27 | 32 | |
| Female | 8 | 5 | |
| ECOG PS | | | 0.661 |
| 0 | 8 | 12 | |
| 1 | 25 | 23 | |
| 2 | 2 | 2 | |
| Smoking status | | | 0.707 |
| Former/Current | 31 | 34 | |
| Never smokers | 4 | 3 | |
| Histology | | | 0.379 |
| Squamous cell carcinoma | 31 | 34 | |
| Adenocarcinoma | 2 | 2 | |
| Other | 2 | 1 | |
| Prior radiotherapy | | | 0.797 |
| Yes | 24 | 27 | |
| No | 11 | 10 | |
| Prior therapies (n) | | | **0.013** |
| 1 | 33 | 26 | |
| $\geq 2$ | 2 | 11 | |
| PNI | | | 0.815 |
| $< 45$ | 21 | 21 | |
| $\geq 45$ | 14 | 16 | |
| GPS | | | 0.495 |
| 0 | 21 | 27 | |
| 1 | 9 | 6 | |
| 2 | 5 | 4 | |
| NLR | | | 0.801 |
| $< 5$ | 25 | 25 | |
| $\geq 5$ | 10 | 12 | |
| Alb | | | 1.00 |
| $< 3.5$ | 5 | 5 | |
| $\geq 3.5$ | 30 | 32 | |
| CRP | | | 0.208 |
| $< 1.0$ | 21 | 28 | |
| $\geq 1.0$ | 14 | 9 | |

## The impact of nutritional status in the nivolumab cohort

In the nivolumab cohort, patients with only one previous therapy (HR = 2.947, $p$ = 0.008), high PNI (HR = 2.725, $p$ = 0.012), low GPS (HR = 2.691, $p$ = 0.016), high albumin (HR = 4.406, $p$ = 0.007), and low CRP (HR = 2.577, $p$ = 0.025) showed significantly better OS (Table 3). Furthermore, when compared with patients in the taxane cohort, patients with good nutritional status in this cohort showed better HRs for OS. Kaplan–Meier analysis showed significant differences in OS according to nutritional status. Patients with high PNI or low GPS showed significantly longer OS than those with low PNI (18.1 vs. 7.6 months, respectively,

**Table 2. Univariate analysis of overall survival (OS) in taxane cohort.**

| Parameter | Category | N | HR | 95% CI | *p*-value |
|---|---|---|---|---|---|
| Age (years) | < 65/≥ 65 | 19/16 | 1.186 | 0.595–2.363 | 0.628 |
| Sex | Male/female | 27/8 | 0.319 | 0.119–0.853 | **0.023** |
| ECOG PS | 0/1, 2 | 8/27 | 1.187 | 0.523–2.695 | 0.682 |
| Previous radiotherapy | No/Yes | 11/24 | 0.479 | 0.218–1.052 | 0.067 |
| Number of previous therapies | 1/≥ 2 | 33/2 | 8.320 | 1.660–41.706 | 0.010 |
| PNI | ≥ 45/< 45 | 14/21 | 1.326 | 0.651–2.703 | 0.437 |
| GPS | 0/1,2 | 21/14 | 1.831 | 0.880–3.806 | 0.105 |
| NLR | < 5/> 5 | 25/10 | 1.501 | 0.681–3.306 | 0.314 |
| Albumin | ≥ 3.5/< 3.5 | 30/5 | 1.733 | 0.657–4.572 | 0.267 |
| CRP | < 1.0/> 1.0 | 21/14 | 1.831 | 0.880–3.806 | 0.105 |

HR, hazard ratio; CI, confidence interval; ECOG PS, Eastern Cooperative Oncology Group performance status; PNI, prognostic nutritional index; GPS, Glasgow prognostic score; NLR, neutrophil-to-lymphocyte ratio; CRP, C-reactive protein.

*p* = 0.009; Fig 3a) or with high GPS (15.5 vs. 4.3 months, respectively, *p* = 0.012; Fig 3b), whereas patients with low NLR tended to correlate with better OS, although the difference was not significant (13.0 vs. 8.1 months, respectively, *p* = 0.312; Fig 3c). PFS analysis showed that low PNI is significantly associated with better outcome (5.1 vs. 1.9 months, respectively, *p* = 0.018; S4 Fig).

## Discussion

In this study, we found that nutritional status was associated with prognosis, especially with nivolumab treatment. We have previously reported that good nutritional status assessed via GPS is associated with better outcomes; the scoring system may be used as a predictor of outcomes in patients with AGC treated with nivolumab. In the present study, where a similar result was observed in AEC patients, demonstrated a positive predictive role of pretreatment

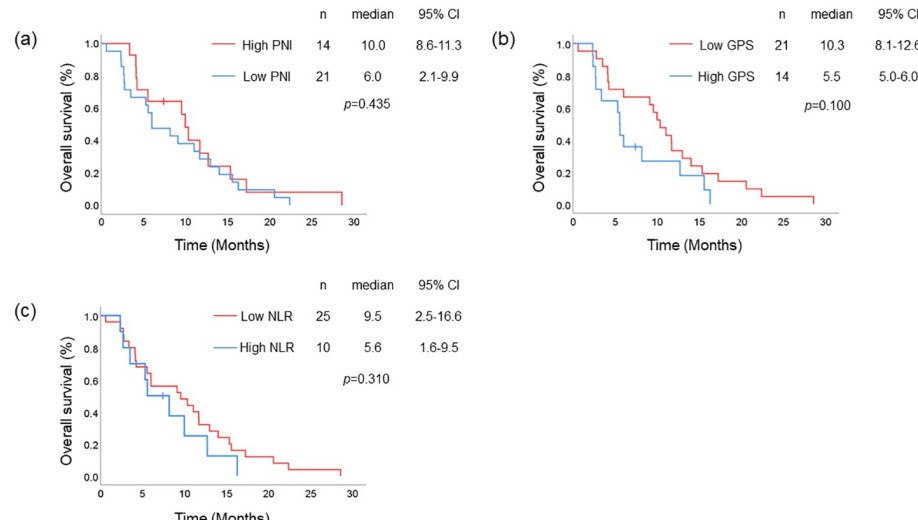

**Fig 2. OS of patients in taxane cohort.** Kaplan–Meier analysis of OS in the taxane cohort: (a) high PNI (red line) and low PNI (blue line), (b) low GPS (red line) and high GPS (blue line), and (c) low NLR (red line) and high NLR (blue line).

**Table 3. Univariate analysis of overall survival (OS) in nivolumab cohort.**

| Parameter | Category | N | HR | 95% CI | *p*-value |
|---|---|---|---|---|---|
| Age (years) | < 65/≥ 65 | 13/24 | 0.765 | 0.347–1.684 | 0.506 |
| Sex | Male/female | 32/5 | 1.125 | 0.427–2.960 | 0.812 |
| ECOG PS | 0/1, 2 | 12/25 | 1.682 | 0.761–3.718 | 0.199 |
| Previous radiotherapy | No/Yes | 10/27 | 1.581 | 0.695–3.594 | 0.274 |
| Number of previous therapies | 1/≥ 2 | 26/11 | 2.947 | 1.332–6.520 | **0.008** |
| PNI | ≥ 45/< 45 | 16/21 | 2.725 | 1.249–5.947 | **0.012** |
| GPS | 0/1,2 | 27/10 | 2.691 | 1.202–6.022 | **0.016** |
| NLR | < 5/> 5 | 25/12 | 1.482 | 0.686–3.202 | 0.317 |
| Albumin | ≥ 3.5/< 3.5 | 32/5 | 4.406 | 1.502–12.928 | **0.007** |
| CRP | < 1.0/> 1.0 | 28/9 | 2.577 | 1.125–5.902 | **0.025** |

HR, hazard ratio; CI, confidence interval; ECOG PS, Eastern Cooperative Oncology Group performance status; PNI, prognostic nutritional index; GPS, Glasgow prognostic score; NLR, neutrophil-to-lymphocyte ratio; CRP, C-reactive protein.

nutritional status on OS in patients with AEC who received immunotherapy over chemotherapy.

Nutritional status may be an independent predictor associated with prognosis in AEC [6,18,19]. Malnutrition is linked to local immunosuppressive environment, such as elevated myeloid-derived suppressor cells (MDSCs), which is associated with poor prognosis in esophageal cancer patients [20]. The systemic nutritional and immunological status of patients has been suggested to affect prognosis through local tumor immunity [6], which suggests the importance of nutritional status for patients treated with nivolumab compared with those treated with chemotherapy. The transition rate of the subsequent therapy may affect patients' clinical outcomes. Patients with good nutritional status may receive subsequent chemotherapy in the nivolumab cohort (10 out of 16 patients with good nutritional status and 11 out of 21

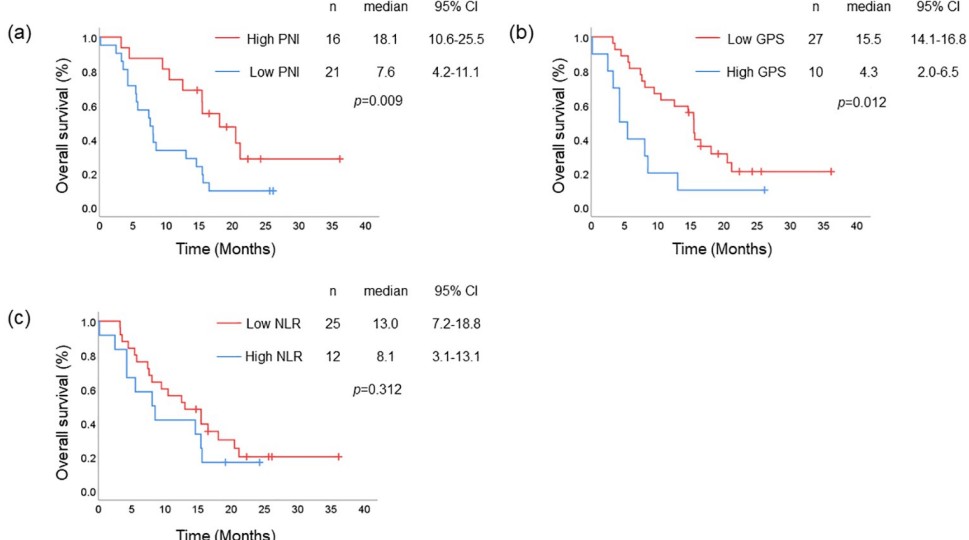

**Fig 3. OS of patients in nivolumab cohort.** Kaplan–Meier analysis of OS in the nivolumab cohort: (a) high PNI (red line) and low PNI (blue line), (b) low GPS (red line) and high GPS (blue line), and (c) low NLR (red line) and high NLR (blue line).

patients with poor nutritional status have received subsequent chemotherapy classified by PNI, and 17 out of 27 patients with good nutritional status and 4 out of 10 patients with poor nutritional status have received subsequent chemotherapy classified by GPS). Another reason for better HRs for OS in the nivolumab cohort when stratified based on nutritional status might be the positive effect of nivolumab on subsequent therapy. The REVIVE study, which prospectively evaluated the efficacy and safety of chemotherapy in nivolumab-refractory or nivolumab-intolerant patients with AGC, reported several additional effects of subsequent chemotherapy [21]. These additional effects may also be observed in AEC, especially in patients with good nutritional status.

We evaluated the association of nivolumab monotherapy with nutritional status, although the combination of immunotherapy and chemotherapy is the standard treatment option for some cancers [22–25]. The phase III checkmate-649 trial [23] evaluated the clinical outcomes of combination therapy with nivolumab and chemotherapy in patients with Combined Positive Score (CPS)-positive gastric cancer and reported the superior effect of combination therapy over chemotherapy alone. An additional effect of nivolumab was not observed in patients with low albumin levels (< lower limit of normal range) [26]. Taken together, previous reports, along with our findings, suggest that good nutritional status is beneficial for patients undergoing immunotherapy.

Biomarkers that can predict the outcome of nivolumab treatment in AEC have been reported [27,28]. A prospective trial has reported that a higher proportion of PD-L1-positive patients had a Complete Response (CR) or Partial Response (PR) compared with PD-L1-negative patients [27]. In AGC, PD-L1 expression, mismatch repair status, and cancer genome alterations have also been suggested as predictive factors [29]. Ikoma et al. have reported that inflammatory prognostic factors were useful in predicting the prognosis for ESCC patients pretreated with nivolumab [30], while Inoue et al. suggested the inflammatory prognostic factors as biomarkers for adverse events and favorable response to nivolumab [31]. The contribution of our study lies in the fact that we also analyzed the effect of nutritional status on the taxane therapy and evaluated the correlation between nutritional status and prognosis of AEC in patients treated with each therapy. Several other predictive biomarkers have been studied; however, there are no methods to modulate these factors. In contrast, the nutritional status of patients can be managed with nutritional interventions [32–34]. The association between nutritional status and clinical outcomes in patients treated with nivolumab has not been prospectively assessed. Therefore, to examine the association, we are currently conducting another prospective clinical trial to explore the relationship between the nutritional status and the efficacy of nivolumab treatment in patients with AEC and AGC (ENTRANCE study; UMIN number: 000043548), and evaluate the efficacy of nivolumab in combination with nutritional counseling in AGC and AEC. In addition, we evaluated QoL, body composition, energy intake, and patients' cellular immunologic profiles using flow cytometry. This study provides novel insights into the association between nutritional status and immunologic profiles of patients.

This study has some limitations. First, we did not analyze the PD-L1 and CD8+ status of tumors and tumor-infiltrating lymphocytes (TILs), which are predictive factors [27]. Second, the retrospective nature of this analysis may have introduced potential bias of confounding factors. Therefore, a further prospective validation study is warranted to evaluate the potential clinical application of the findings. Third, different effects are observed in squamous cell carcinoma and adenocarcinoma of AEC patients treated with immune checkpoint inhibitors [35], and subgroup analysis on histological types could not be conducted due to the small sample size. Lastly, although the nutritional status is a prognostic factor for survival in patients with AEC, taxane therapy showed no differences. The small sample size might have affected the

results; however, when compared with those in the taxane cohort, patients with good nutritional status in the nivolumab cohort showed better hazard ratios for OS.

In conclusion, pretreatment nutritional status is an important factor for prognosis, especially in patients treated with nivolumab, and a potential predictor of nivolumab treatment efficacy in patients with AEC. Our findings suggest that nutritional management prior to nivolumab therapy is essential to achieve better outcomes from immune therapies.

## Supporting information

**S1 Fig. PFS and OS of patients in taxane cohort.** Kaplan–Meier analysis of (a) PFS and (b) OS.
(TIF)

**S2 Fig. PFS and OS of patients in nivolumab cohort.** Kaplan–Meier analysis of (a) PFS and (b) OS.
(TIF)

**S3 Fig. PFS of patients in taxane cohort.** Kaplan–Meier analysis of PFS in the taxane cohort: (a) high PNI (red line) and low PNI (blue line), (b) low GPS (red line) and high GPS (blue line), and (c) low NLR (red line) and high NLR (blue line).
(TIF)

**S4 Fig. PFS of patients in nivolumab cohort.** Kaplan–Meier analysis of PFS in the nivolumab cohort: (a) high PNI (red line) and low PNI (blue line), (b) low GPS (red line) and high GPS (blue line), and (c) low NLR (red line) and high NLR (blue line).
(TIF)

**S1 Table. Overall response rate in the taxane cohort and in the nivolumab cohort.**
(TIF)

## Acknowledgments

We wish to thank all staff involved in esophageal cancer treatment.

## Author Contributions

**Conceptualization:** Naoki Takegawa, Masahiro Tsuda.

**Data curation:** Naoki Takegawa.

**Formal analysis:** Naoki Takegawa.

**Investigation:** Naoki Takegawa, Taku Hirabayashi, Shunta Tanaka, Michiko Nishikawa, Nagahiro Tokuyama, Takuya Mimura, Saeko Kushida, Hidetaka Tsumura, Yoshinobu Yamamoto, Ikuya Miki, Masahiro Tsuda.

**Methodology:** Naoki Takegawa, Masahiro Tsuda.

**Project administration:** Masahiro Tsuda.

**Supervision:** Masahiro Tsuda.

**Writing – original draft:** Naoki Takegawa.

**Writing – review & editing:** Michiko Nishikawa, Nagahiro Tokuyama, Takuya Mimura, Hidetaka Tsumura, Yoshinobu Yamamoto, Masahiro Tsuda.

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
