## [Decision Letter · Decision Letter 0]

16 Feb 2023

PONE-D-22-30499The impact of nutritional status in nivolumab-treated patients with advanced esophageal cancerPLOS ONE

Dear Dr. tsuda,

Thank you for submitting your manuscript to PLOS ONE. After careful consideration, we feel that it has merit but does not fully meet PLOS ONE’s publication criteria as it currently stands. Therefore, we invite you to submit a revised version of the manuscript that addresses the points raised during the review process.

We look forward to receiving your revised manuscript.

Kind regards,

Kenji Fujiwara, PhD, MD

Academic Editor

PLOS ONE

Journal Requirements:

Additional Editor Comments:

Dear Dr. Tsuda.

I inherited this manuscript from another Academic Editor. I feel sorry for you for the long period of the review process. The manuscript is about the contribution of nutritional status for the treatment of esophageal cancer with an immune checkpoint inhibitor. The results show the statistical significance and I think the data sounds intriguing to many readers. I agree to two reviewers and this manuscript is eligible to proceed to major revision.

I added my own comments below. I look forward to seeing the revision.

Best,

Kenji Fujiwara

Major concerns.

1. The results seem interesting but the numbers of patients are limited, especially in the Nivolumab group. I think the authors may be able to add more patients now. Please update it, if possible.

2. It seems rational to me that nutritional status affects the prognosis but Taxane did not show the difference. This might be the question for many readers. Could you discuss the irrelevance of nutritional status in chemotherapy in the Discussion section?

3. I agree with a reviewer about the necessity of English proofreading.

4. For the ethics statement, I understand it is difficult to get consent in a retrospective study like this. However, the authors' statement “Consent was not obtained because the data were analyzed anonymously," sounds irrational because most studies are analyzed anonymously. Please change the description. At least, the authors should describe that the ethics committee waived the requirement for informed consent.

5. From the author contributions, only two authors ran the study and nine authors did only writing–review and editing. I am surprised that only one person ran the analysis and investigation, but the contributions seem unbalanced. Please reconfirm the contributions by checking the guidelines.

https://journals.plos.org/plosone/s/authorship#loc-author-contributions

6. Introduction consists of too many paragraphs. Please reconstruct it.

7. Whether Squamous cell carcinoma or adenocarcinoma is a very interesting point in the effectiveness of treatment. The authors may add the data analyzing the prognosis separately by depending on histology or at least discuss it.

Minor concerns.

1. The names of persons and addresses are not written in capital letters sometimes in the submission system. Please adjust them.

2. Please show elaboration for the abbreviation when it first appeared. For example, “AEC” was used without description in the abstract.

3. I think PLOS ONE does not ask the authors to include a short title and authors' contribution in the title pages. Please refer to the guidelines and adjust them.

https://journals.plos.org/plosone/s/submission-guidelines

Reviewers' comments:

Reviewer's Responses to Questions

**Comments to the Author**

1. Is the manuscript technically sound, and do the data support the conclusions?

Reviewer #1: Partly

Reviewer #2: Yes

2. Has the statistical analysis been performed appropriately and rigorously? 

Reviewer #1: Yes

Reviewer #2: Yes

3. Have the authors made all data underlying the findings in their manuscript fully available?

Reviewer #1: Yes

Reviewer #2: Yes

4. Is the manuscript presented in an intelligible fashion and written in standard English?

Reviewer #1: Yes

Reviewer #2: Yes

5. Review Comments to the Author

Reviewer #1: Dear Sir:

I have read the manuscript entitled "The impact of nutritional status in nivolumab-treated patients with advanced esophageal cancer". The study explored whether there is a correlation between nutritional status (Glasgow prognostic score, prognostic nutritional index, and neutrophil-to-lymphocyte ratio) and prognosis of advanced esophageal cancer (AEC) in patients treated with taxane or nivolumab therapy. In the nivolumab cohort (N=28), patients with good nutritional status had significantly better median overall survival than those with poor nutritional status (15.5 vs. 7.6 months, respectively, p=0.017, classified by prognostic nutritional index, 14.6 vs. 5.5 months, respectively, p=0.022, classified by Glasgow prognostic score), whereas the prognosis of the patients treated with taxane therapy was less affected by the nutritional status. This observation suggests that pretreatment nutritional status of patients with advanced esophageal cancer is a key factor for successful outcomes, especially for nivolumab treatment.

I have several comments:

1.English editing is necessary

2.As stated by the authors, the retrospective nature of this analysis and limited number of cases may have introduced potential bias of confounding factors. Also both patients with squamous cell carcinoma and adenocarcinoma were included. These two different types of histology may further bias the result. Authors should discuss this point.

3.OS could be influenced by subsequent therapy. Is it possible that in nivolumab group, patients with relatively good nutrition status may have higher chance to receive subsequent chemotherapy? On the contrary, is it possible that in taxane group, patients with either good or poor nutritional status could have similar chance to receive nivolumab after taxane due to relatively low toxicity of immunotherapy? Could authors provide information regarding subsequent therapy in both groups following failure of nivolumab or taxane? Authors should also discuss this possibility in Discussion section.

4.Is there any papers discussing about the association between nutritional status and other biomarkers predicting response to immunotherapy, such as PD-L1 expression, T cell population, inflammation gene signature, etc.? Please discuss

5.Could authors release more detailed information about the design of ENTRANCE study and explain how this study could explore this question?

Reviewer #2: Thank you for your valuable consideration.

Although the number of cases is small, it is interesting to compare the data between taxane and nivolumab regarding the relationship between nutritional status and therapeutic effect.

6. PLOS authors have the option to publish the peer review history of their article (what does this mean?). If published, this will include your full peer review and any attached files.

Reviewer #1: **Yes: **Chueh-Chuan Yen

Reviewer #2: No

---

## [Author Response · Author response to Decision Letter 0]

31 Mar 2023

Response to comments from reviewers and the associated editor

Thank you for your comments and suggestions. Kindly see our replays to the comments raised below

Response: We have ensured that the manuscript adheres to the formatting requirements of PLOS One

Response: We have provided this information in the revised manuscript (page 5, lines 102–103.)

Response: This study received no specific funding for this work. We have incorporated the statements within cover letter.

Additional Editor Comments:

Dear Dr. Tsuda.

I inherited this manuscript from another Academic Editor. I feel sorry for you for the long period of the review process. The manuscript is about the contribution of nutritional status for the treatment of esophageal cancer with an immune checkpoint inhibitor. The results show the statistical significance and I think the data sounds intriguing to many readers. I agree to two reviewers and this manuscript is eligible to proceed to major revision.

I added my own comments below. I look forward to seeing the revision.

Best,

Kenji Fujiwara

Major concerns.

1. The results seem interesting but the numbers of patients are limited, especially in the Nivolumab group. I think the authors may be able to add more patients now. Please update it, if possible.

Response: Thank you for your suggestion. Nine additional patients were analyzed, and all patients were followed up until March 1, 2023.

2. It seems rational to me that nutritional status affects the prognosis but Taxane did not show the difference. This might be the question for many readers. Could you discuss the irrelevance of nutritional status in chemotherapy in the Discussion section?

Response: Thank you for your suggestion. Indeed, nutritional status is a prognostic factor for survival in patients with AEC. In the taxane cohort, patients with good nutritional status tended to have longer OS (HR=1.326, p=0.437 [classified by PNI]; HR=1.831, p=0.105 [classified by GPS]; HR=1.501, p=0.314 [classified by NLR]), but this increase was not statistically significant. The small sample size might have affected the results. However, when compared with those in the taxane cohort, patients with good nutritional status in the nivolumab cohort showed better hazard ratios for OS. We have discussed this in the revised Discussion section (page 17, lines 301–305.).

3. I agree with a reviewer about the necessity of English proofreading.

Response: Thank you for your suggestion. The manuscript was revised by a native English-speaking editor.

4. For the ethics statement, I understand it is difficult to get consent in a retrospective study like this. However, the authors' statement “Consent was not obtained because the data were analyzed anonymously," sounds irrational because most studies are analyzed anonymously. Please change the description. At least, the authors should describe that the ethics committee waived the requirement for informed consent.

Response: Thank you for your suggestion. We have revised the text accordingly (page 5, lines 102–103.).

5. From the author contributions, only two authors ran the study and nine authors did only writing–review and editing. I am surprised that only one person ran the analysis and investigation, but the contributions seem unbalanced. Please reconfirm the contributions by checking the guidelines.

https://journals.plos.org/plosone/s/authorship#loc-author-contributions

Response: We have reconfirmed the contributions.

6. Introduction consists of too many paragraphs. Please reconstruct it.

Response: Thank you for your suggestions. We have reconstructed the Introduction section.

7. Whether Squamous cell carcinoma or adenocarcinoma is a very interesting point in the effectiveness of treatment. The authors may add the data analyzing the prognosis separately by depending on histology or at least discuss it.

Response: Thank you for your suggestion. Indeed, different effects are observed in squamous cell carcinoma and adenocarcinoma of AEC patients treated with immune checkpoint inhibitors. In the present study, SCC patients with good nutritional status had significantly better median overall survival than those with poor nutritional status (15.4 vs. 4.3 months, respectively, p=0.017, classified by PNI, 15.5 vs. 4.3 months, respectively, p=0.026, classified by GPS). We did not show the data in the current manuscript but described this point in the revised Discussion section (page 17, lines 298–301.). Patients with non-squamous cell carcinoma were not analyzed due to the small sample size.

Minor concerns.

1. The names of persons and addresses are not written in capital letters sometimes in the submission system. Please adjust them.

Response: We have adjusted them in the submission system.

2. Please show elaboration for the abbreviation when it first appeared. For example, “AEC” was used without description in the abstract.

Response: We have defined all abbreviations when first appeared. We have removed abbreviations in the abstract.

Thank you for your valuable consideration.

Although the number of cases is small, it is interesting to compare the data between taxane and nivolumab regarding the relationship between nutritional status and therapeutic effect.

Comment 1.

There is a similar study with a large number of patients on nivolumab, so please cite and consider the content.( Inflammatory prognostic factors in advanced or recurrent esophageal

squamous cell carcinoma treated with nivolumab, Cancer Immunology, Immunotherapy

https://doi.org/10.1007/s00262-022-03265-7

Response: Thank you for your suggestion. We have cited the study. The contributions of our study lie in the fact that we analyzed the effect of nutritional status on taxane therapy.

Comment 2.

Enter the number of patients after each cohort in the first row of table.1.

Response: Thank you for your suggestion. We have stated the number of patients after each cohort in Table 1.

3. I think PLOS ONE does not ask the authors to include a short title and authors' contribution in the title pages. Please refer to the guidelines and adjust them.

https://journals.plos.org/plosone/s/submission-guidelines

Response: We have removed the short title and authors' contribution from the title page. A short title is not required?

Reviewers' comments:

Reviewer's Responses to Questions

Comments to the Author

1. Is the manuscript technically sound, and do the data support the conclusions?

Reviewer #1: Partly

Reviewer #2: Yes

2. Has the statistical analysis been performed appropriately and rigorously?

Reviewer #1: Yes

Reviewer #2: Yes

3. Have the authors made all data underlying the findings in their manuscript fully available?

Reviewer #1: Yes

Reviewer #2: Yes

4. Is the manuscript presented in an intelligible fashion and written in standard English?

Reviewer #1: Yes

Reviewer #2: Yes

5. Review Comments to the Author

Reviewer #1: Dear Sir:

I have read the manuscript entitled "The impact of nutritional status in nivolumab-treated patients with advanced esophageal cancer". The study explored whether there is a correlation between nutritional status (Glasgow prognostic score, prognostic nutritional index, and neutrophil-to-lymphocyte ratio) and prognosis of advanced esophageal cancer (AEC) in patients treated with taxane or nivolumab therapy. In the nivolumab cohort (N=28), patients with good nutritional status had significantly better median overall survival than those with poor nutritional status (15.5 vs. 7.6 months, respectively, p=0.017, classified by prognostic nutritional index, 14.6 vs. 5.5 months, respectively, p=0.022, classified by Glasgow prognostic score), whereas the prognosis of the patients treated with taxane therapy was less affected by the nutritional status. This observation suggests that pretreatment nutritional status of patients with advanced esophageal cancer is a key factor for successful outcomes, especially for nivolumab treatment.

I have several comments:

1. English editing is necessary

Response: Thank you for your suggestion. Manuscript has been edited by native English-speaking editor.

2. As stated by the authors, the retrospective nature of this analysis and limited number of cases may have introduced potential bias of confounding factors. Also both patients with squamous cell carcinoma and adenocarcinoma were included. These two different types of histology may further bias the result. Authors should discuss this point.

Response: Thank you for your suggestion. Indeed, different effects are observed in squamous cell carcinoma and adenocarcinoma of AEC patients treated with immune checkpoint inhibitors. In the present study, SCC patients with good nutritional status had significantly better median overall survival than those with poor nutritional status (15.4 vs. 4.3 months, respectively, p=0.017, classified by PNI, 15.5 vs. 4.3 months, respectively, p=0.026, classified by GPS). We did not show the data in the current manuscript but described this point in the revised Discussion section (page 17, lines 298–301.). Patients with non-squamous cell carcinoma were not analyzed due to the small sample size.

3.OS could be influenced by subsequent therapy. Is it possible that in nivolumab group, patients with relatively good nutrition status may have higher chance to receive subsequent chemotherapy? On the contrary, is it possible that in taxane group, patients with either good or poor nutritional status could have similar chance to receive nivolumab after taxane due to relatively low toxicity of immunotherapy? Could authors provide information regarding subsequent therapy in both groups following failure of nivolumab or taxane? Authors should also discuss this possibility in Discussion section.

Response: Thank you for your suggestion. The overall survival could be influenced by subsequent therapy. Patients with good nutritional status may receive subsequent chemotherapy in the nivolumab cohort (10 out of 16 patients with good nutritional status and 11 out of 21 patients with poor nutritional status have received subsequent chemotherapy classified by PNI, and 17 out of 27 patients with good nutritional status and 4 of 10 patients with poor nutritional status have received subsequent chemotherapy classified by GPS). In the taxane cohort, only 1 patient received subsequent systemic chemotherapy with docetaxel. We discuss this point in the revised Discussion section (pages 14–15, lines 248–254.).

4.Is there any papers discussing about the association between nutritional status and other biomarkers predicting response to immunotherapy, such as PD-L1 expression, T cell population, inflammation gene signature, etc.? Please discuss.

Response: Thank you for your suggestion. We have cited other papers discussing other biomarkers that predict responses to immunotherapy in AEC. The contribution of our study lies in the fact that we analyzed the effect of nutritional status on taxane therapy.

5.Could authors release more detailed information about the design of ENTRANCE study and explain how this study could explore this question?

Response: The ENTRANCE study is an observational study that evaluates the efficacy of nivolumab in combination with nutritional counseling in AGC and AEC. In addition, we evaluated QoL, body composition, energy intake, and patients’ cellular immunologic profiles using flow cytometry. This study provides novel insights into the association between nutritional status and immunologic profiles of patients. We have added information on the design of the ENTRANCE study to the revised manuscript (page 16, lines 286–293.).

Reviewer #2: Thank you for your valuable consideration.

Although the number of cases is small, it is interesting to compare the data between taxane and nivolumab regarding the relationship between nutritional status and therapeutic effect.

6. PLOS authors have the option to publish the peer review history of their article (what does this mean?). If published, this will include your full peer review and any attached files.

Do you want your identity to be public for this peer review? For information about this choice, including consent withdrawal, please see our Privacy Policy.

Reviewer #1: Yes: Chueh-Chuan Yen

Reviewer #2: No

---

## [Decision Letter · Decision Letter 1]

24 Apr 2023

The impact of nutritional status in nivolumab-treated patients with advanced esophageal cancer

PONE-D-22-30499R1

Dear Dr. tsuda,

We’re pleased to inform you that your manuscript has been judged scientifically suitable for publication and will be formally accepted for publication once it meets all outstanding technical requirements.

Kind regards,

Kenji Fujiwara, PhD, MD

Academic Editor

PLOS ONE

Additional Editor Comments (optional):

Dear Dr. Tsuda.

Thank you for revising your manuscript appropriately. All reviewers and I agreed to the acceptance.

Yours sincerely,

Kenji Fujiwara

Academic editor

Reviewers' comments:

Reviewer's Responses to Questions

**Comments to the Author**

1. If the authors have adequately addressed your comments raised in a previous round of review and you feel that this manuscript is now acceptable for publication, you may indicate that here to bypass the “Comments to the Author” section, enter your conflict of interest statement in the “Confidential to Editor” section, and submit your "Accept" recommendation.

Reviewer #1: All comments have been addressed

Reviewer #2: All comments have been addressed

2. Is the manuscript technically sound, and do the data support the conclusions?

Reviewer #1: Yes

Reviewer #2: Yes

3. Has the statistical analysis been performed appropriately and rigorously? 

Reviewer #1: Yes

Reviewer #2: Yes

4. Have the authors made all data underlying the findings in their manuscript fully available?

Reviewer #1: Yes

Reviewer #2: Yes

5. Is the manuscript presented in an intelligible fashion and written in standard English?

Reviewer #1: Yes

Reviewer #2: (No Response)

6. Review Comments to the Author

Reviewer #1: As stated in my previous comment, this is a study with small sample size. So the information that could be given is limited. However, in the response letter, authors have provided a well point-to-point response for the comments. Hopefully authors could explore relevant issue in the future study with a larger sample size.

Reviewer #2: Thank you for the opportunity to peer review this wonderful study.

You have answered all the comments properly and I have nothing to point out.

7. PLOS authors have the option to publish the peer review history of their article (what does this mean?). If published, this will include your full peer review and any attached files.

Reviewer #1: **Yes: **Chueh-Chuan Yen

Reviewer #2: **Yes: **Toshihiko Matsumoto

---

## [Editor Report · Acceptance letter]

27 Apr 2023

PONE-D-22-30499R1 

The impact of nutritional status in nivolumab-treated patients with advanced esophageal cancer 

Dear Dr. Tsuda:

I'm pleased to inform you that your manuscript has been deemed suitable for publication in PLOS ONE. Congratulations! Your manuscript is now with our production department. 

Kind regards, 

on behalf of

Dr. Kenji Fujiwara 

Academic Editor

PLOS ONE